# The Inorganic Perovskite-Catalyzed Transfer Hydrogenation of Cinnamaldehyde Using Glycerol as a Hydrogen Donor

Tafadzwa Precious Mabate [1], Reinout Meijboom [1,2] and Ndzondelelo Bingwa [1,2,*]

1   Research Center for Synthesis and Catalysis, Department of Chemical Sciences, University of Johannesburg, P.O. Box 524, Auckland Park 2006, South Africa; 201516781@student.uj.ac.za (T.P.M.); rmeijboom@uj.ac.za (R.M.)
2   Centre for Nanomaterials Science Research, University of Johannesburg, Johannesburg 2092, South Africa
*   Correspondence: nbingwa@uj.ac.za; Tel.: +27-115-592-363; Fax: +27-115-592-819

**Abstract:** Catalytic transfer hydrogenation reactions (CTHs) produce value-added chemicals in the most economical, safe, green, and sustainable way. However, understanding the reaction mechanism and developing stable, selective, and cheap catalysts has been a significant challenge. Herein, we report on the hydrogenation of cinnamaldehyde utilizing glycerol as a hydrogen donor and metal-oxides ($SnO_2$, $LaFeO_3$, and $LaSnO_3$) as heterogeneous catalysts. The perovskite types were used because they are easy to synthesize, the metal components are readily available, and they are good alternatives to noble metals. The catalysts were synthesized through the nanocasting (hard-template) method with $SiO_2$ (KIT-6) as a template. The template was synthesized using the soft-template (sol-gel) method resulting in a high surface area of 624 $m^2/g$. Furthermore, catalytic evaluations gave high cinnamaldehyde percentage conversions of up to 99%. Interestingly, these catalysts were also found to catalyze the etherification of glycerol in one pot. Therefore, we propose competitive surface catalytic reactions driven by the transition metal cations as the binding sites for the cinnamaldehyde and the sacrificial glycerol.

**Keywords:** catalytic transfer hydrogenation; glycerol; etherification; perovskites

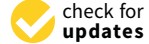



## 1. Introduction

Sustainability is vital in different reactions as this provides the means to meet human needs using the efficiency of natural products for chemicals and services. Several reactions are performed to increase sustainability, such as converting biomass to fuels using renewable chemicals [1–6]. The catalytic transfer hydrogenation reaction (CTH) also follows that direction as it uses renewable hydrogen donors such as bioderived sacrificial alcohols [7]. Moreover, this is a green approach that reduces greenhouse emissions and the pollution of the environment. The replacement of molecular $H_2$ with biomass-derived hydrogen donors also provides a safer alternative, as molecular hydrogen requires specialized handling and transportation and is deemed to be unsafe [8–10]. Furthermore, the same phase of the hydrogen donor and substrate increases the contact time, thus enhancing reaction efficiency due to the transport phenomena [11].

Previous reactions have used biomass-derived hydrogen donors for CTHs. Isopropanol and other monoprotic alcohols are the most used [12]. Among the monoprotic donors, isopropanol is a better hydrogen donor than n-propanol, ethanol, and methanol [10]. However, very few studies have been conducted using glycerol, a polyprotic sacrificial alcohol [11,13,14]. Using glycerol instead of monoprotic donors could be a practical approach because it can also be used as a solvent.

On compounds that have more than one reduction site in CTHs, selectivity for the desired product becomes a challenge. For example, in α,β-unsaturated combinations, two sites can be hydrogenated. These are the alkene double bond and the terminal carbonyl of the aldehyde moiety [15]. These two sites make it particularly challenging to synthesize

catalysts that are selective to one desired product. Several catalysts have been synthesized and used for this reaction, including nitrogen-doped, carbon-supported iron [10], Pd-based catalysts [15], Ir/C catalysts [16], RANEY Ni [9], platinum oxide, platinum black catalysts, and promoters [17]. In particular, the carbon-supported iron catalysts were stable for the CTH of furfural-to-furfural alcohol with conversions greater than 90% under optimum conditions. However, the presence of crystalline oxide materials together with pore structure variation led to catalyst deactivation [10,18–20].

Several transition metal catalysts have been employed in CTH reactions and have led to conversions of different substrates such as arenes, $\alpha,\beta$-unsaturated compounds, aldehydes, and ketones. Some transition metals have been used in bimetallic systems. In the study conducted by Xiang et al., the complete conversion of styrene and nitrobenzene using methanol as a hydrogen donor was achieved, with good conversions dependent on the hydrogen donor water ratio [21]. However, relatively large amounts of catalysts were required, therefore rendering the reaction expensive. This study, amongst others, motivates the urge to develop cheap, nonprecious metal-based catalytic systems that utilize abundant active metals [10].

Herein, the use of Sn and Fe-based perovskites (LaFeO$_3$ and LaSnO$_3$) as catalysts in the catalytic transfer hydrogenation of cinnamaldehyde using glycerol as a hydrogen donor is reported. Perovskites are thermally and hydrothermally stable integral mixed oxides that do not easily leach out of the reaction like traditional catalysts for CTH reactions [11,22,23]. They have the formula ABO$_3$ and can be used for various reactions and applications due to their electrochemical properties and porosity [24]. They have also been utilized in different applications such as fuel cells, the purification of automobile exhausts, the decomposition of N$_2$O, reactions involving water gas shifts, photocatalytic water splitting, and chemical looping combustion reactions [25–30]. To the best of our knowledge, very few studies have reported using perovskites in CTHs. In these few studies, perovskites have been used in CTH reactions using isopropanol or methanol as a hydrogen donor, and not the abundant glycerol [7,21,31]. A recent study has been published that explored the conversion of furfural-to-furfural alcohol using carbon composites and perovskites. This provided an alternative route to noble metal catalysts [11,32]. Scheme 1 shows possible cinnamaldehyde hydrogenation pathways from cinnamaldehyde (1) to cinnamyl alcohol (2) [33,34], hydrocinnamaldehyde (3), [33,35] and the further hydrogenation of (1) and (2) to form phenyl propanol (4) [36,37].

Studies have also been reported on tuning the acid-base properties of the catalysts to affect the reaction's progress. It has been proven that acidity affects selectivity in CTH reactions, whereas basicity affects the percentage conversion of the catalysts [11]. The acid-base properties of perovskites are easily tunable due to the possibility of metal substitution on the B site of the ABO$_3$ system. This tunability leads to catalysts, where fine-tuning of the redox properties is possible and is earmarked as alternatives to the depletion of expensive precious metals. Furthermore, cinnamaldehyde occurs naturally in cinnamon bark and can be converted into numerous chemical products with many applications. This study aims to hydrogenate cinnamaldehyde into cinnamyl alcohol using Sn and Fe-based perovskite as heterogeneous catalysts.

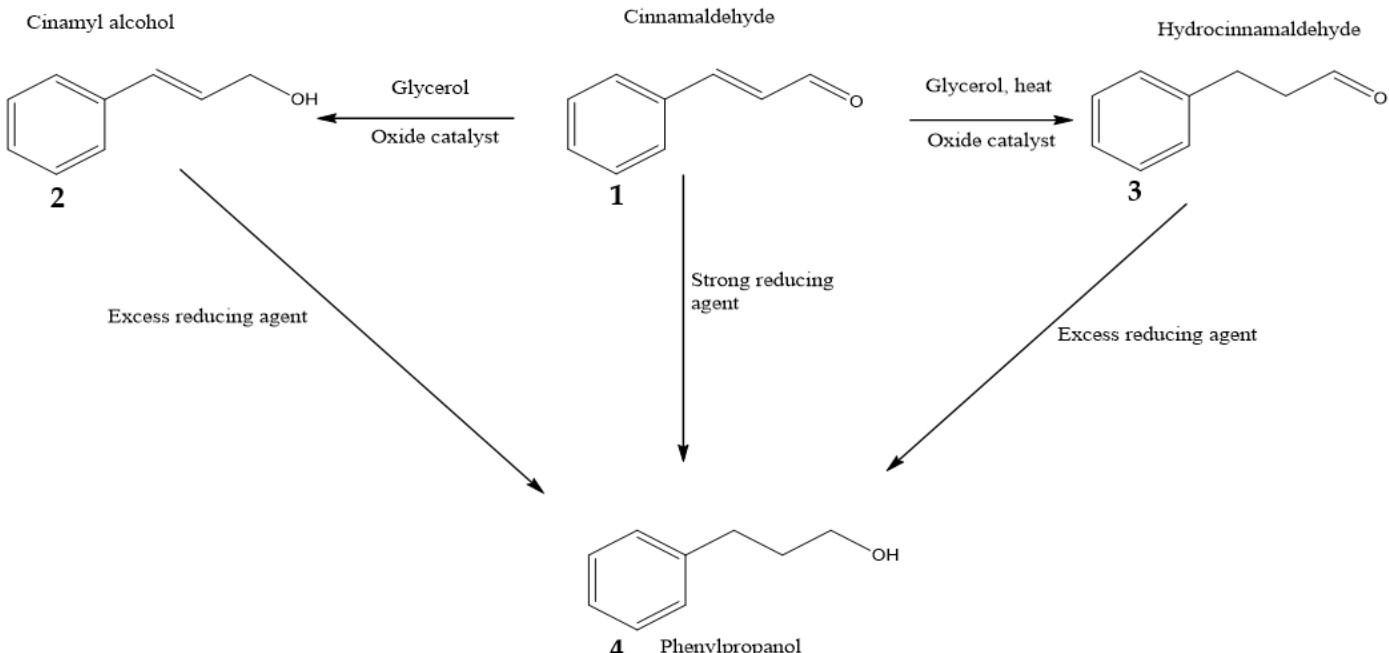

**Scheme 1.** Hydrogenation of cinnamaldehyde to various products in the presence of oxide catalysts.

## 2. Results and Discussions

### 2.1. Catalyst's Characterization

Figure 1 shows the p-XRD patterns that correspond to the $SnO_2$ and perovskite $ABO_3$ lattice structures. These results show the main phases of $SnO_2$, $LaFeO_3$, and $LaSnO_3$, proving the successful synthesis of these materials. The $LaFeO_3$ perovskite pattern corresponds to the perovskite structure reported by Xiao et al., with the characteristic peak (CK) around 32.5° (2θ) [11]. Herein, the peak at 32.2° (2θ) corresponds to the (121) crystal face. In addition, none of the peaks observed were assignable to the individual La or iron oxide, which signifies the existence of the solid material produced to be perovskite phases only [11]. For the $LaSnO_3$ perovskite, the characteristic peak (CK) appeared at 33° (2θ), which proved the presence of the perovskite lattice. The perovskites had patterns that correspond to the perovskite structure reported by Xiao et al. [11]. The peaks observed correspond to the Miller indices shown in Figure 1 and are like those reported in previous studies [38–40]. Reference cards with JCPD numbers 04-002-0289, 04-055-6880, and 01-088-0641 were obtained for $SnO_2$, $LaSnO_3$, and $LaFeO_3$, respectively from the High Score Plus software. Peaks around 25–27° (2θ) in the diffractograms indicate residual silica from the synthesis procedure.

The porous structures of the KIT-6 template and the catalysts were confirmed using the $N_2$ adsorption–desorption measurements, as illustrated in Figure 2. All catalysts showed type IV isotherms with hysteresis loops, typical of mesoporous materials. The BET surface areas, pore volumes, and pore diameters are summarised in Table 1. All materials had a narrow pore size distribution showing the uniformity of the porous structure of these materials. These pore sizes range from 4 to 13.1 nm, indicating the mesoporous range [15]. The KIT-6 was found to have a high surface area of 625 $m^2/g$. This value is in the range of other KIT-6 materials previously reported [41,42]. The perovskites also have large and acceptable surface areas ranging from 135 to 687 $m^2/g$, typical of nanocast materials.

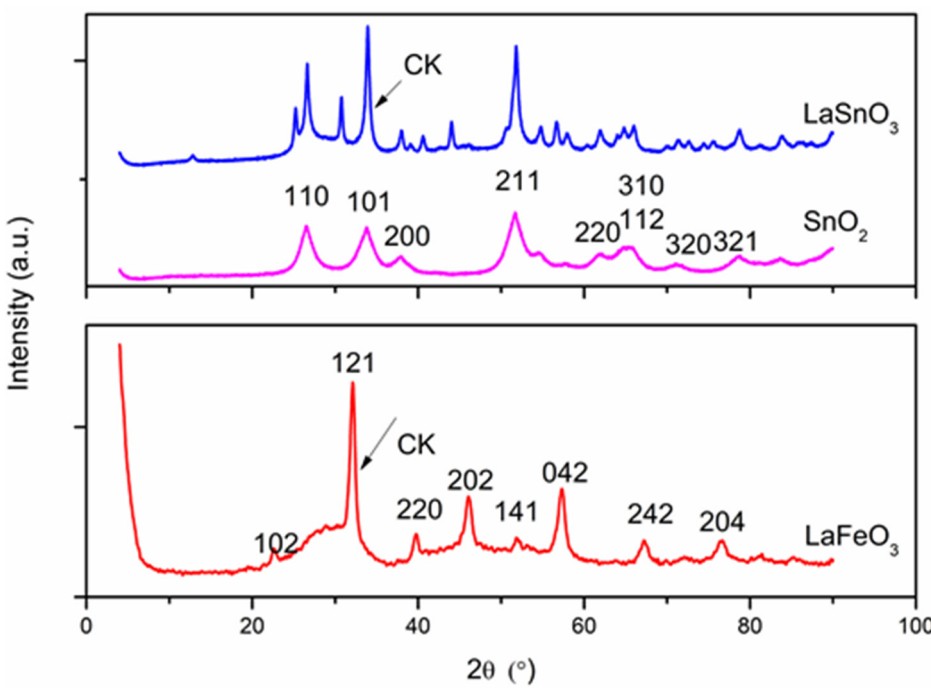

**Figure 1.** XRD patterns of LaFeO$_3$, LaSnO$_3$ (vertically translated by 5661.3 a.u.), and SnO$_2$.

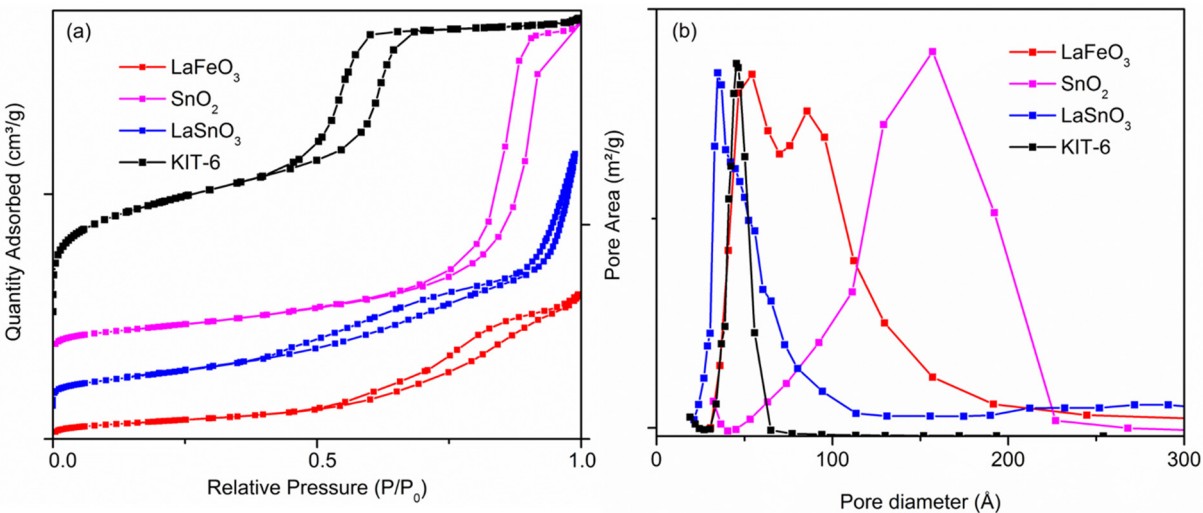

**Figure 2.** Nitrogen sorption isotherms for the synthesized catalyst materials (**a**) and their corresponding pore size distributions (**b**). The LaSnO$_3$ isotherm was vertically translated by 214.7 cm$^3$/g and that for KIT-6 by −3.6 cm$^3$/g.

**Table 1.** Summary of nitrogen sorption measurements of the synthesized catalysts and comparison with the literature.

| Entry | Catalyst | Surface Area (m²/g) | Pore Volume (cm³/g) | Pore Diameter (nm) | Reference |
|---|---|---|---|---|---|
| 1 | KIT-6 (SiO$_2$) | 625 | 0.67 | 4.3 | |
| 2 | LaFeO$_3$ | 135 | 0.37 | 11.0 | This work |
| 3 | LaSnO$_3$ | 687 | 1.91 | 9.9 | |
| 4 | SnO$_2$ | 32 | 0.11 | 13.1 | |
| 5 | KIT-6 (SiO$_2$) | 772 | 0.74 | 5.2 | [15] |
| 7 | LaFeO$_3$ | 92 | 0.33 | 7.7 | [42] |
| 8 | SnO$_2$ | 50 | 0.06 | 5.3 | [43] |

Analysis with microscopes was utilized to further confirm that the synthesized materials are porous. The TEM and SEM were used to evaluate the porous network and morphology of the materials. The TEM results show that the catalysts have porous channels that resemble the pores of the KIT-6 template, as shown in Figure 3a,b. The $SnO_2$ appears as small particulates, as shown in Figure 3c. It is important to note that, unlike the $LaFeO_3$ and $LaSnO_3$, which were prepared via the nanocasting method, $SnO_2$ was prepared using the sol-gel technique, hence the striking difference in structural properties.

The SEM analysis revealed that perovskites have spherical shapes more dominantly than $SnO_2$, which has various shapes. It was also observed that the materials had pores on their surfaces which further confirmed their porous nature. These results are clearly shown in Figure 3d–f. The EDS results, shown in Figure 3j–l, complement the p-XRD results, as La, Fe, Sn, and O cations were the only atoms that formed the catalysts. These were also in line with the SEM elemental mapping results shown in Figure 3g–i. Both the XRD and EDS showed the existence of silica. This minute amount of silica derives from the template used in nanocasting. The Cl comes from the stannous chloride precursor. It was also noted that there were Na elements picked up on the perovskite catalysts from elemental mapping. These Na elements are attributed to the sodium hydroxide that was used to wash the template. The amount of these ions is insignificant and does not alter the catalytic activities of the catalysts.

All the as-synthesized perovskites showed relatively good thermal stability. These materials were thermally stable, showing no fragmentation patterns below 500 °C. The significant loss of weight is the loss of strongly adsorbed water molecules from 100 °C. In the perovskite samples, the degradation at 500 °C is attributed to the removal of the residual hydroxyl group from the decomposition of citric acid ($C_6H_8O_7$). In contrast, the destruction of the $SnO_2$ material is seen as early as 400 °C (see Figure 4a).

On the other hand, the chemical properties of the catalysts were compared to the activity and selectivity trends in the acid-base properties of the catalyst. The distribution of basic sites appears similar for all the catalysts (Figure 4b and Figure S2 in ESI). All the catalysts show relatively small amounts of weak basic sites. At the same time, they also possess a considerable amount of strong basic sites shown by the desorption of carbon dioxide at lower and higher temperatures, respectively. The basicity of the catalysts is tabulated in Table 2.

**Table 2.** A catalyst comparison study of the prepared perovskites with previously reported catalysts.

| Entry | Catalyst | Reaction Time (h) | Basicity (a.u) | Temperature °C | % Conv. [a] | HCAL % Selec. [b] | Reference |
|---|---|---|---|---|---|---|---|
| 1 | $LaFeO_3$ | 6 | 0.369 | 180 | 99.0 | 82.1 | This work |
| 2 | $LaSnO_3$ | 6 | 0.636 | 180 | 99.3 | 4.9 | |
| 3 | $SnO_2$ | 6 | 0.324 | 180 | 98.7 | 22.1 | |
| 4 | $LaFeO_3$ | 3 | - | 80 | 21.8 | 100 | [15] |
| 5 | $Ru_{0.05}Sn_{0.25}Ti_{0.7}O_2/Ti$ | - | - | 30 | 86.2 | 19.3 | [44] |

[a] Conversion $= \frac{\text{moles of cinnamaldehyde consumed}}{\text{moles of cinnamldehyde in the feed}} \times 100$, [b] Selectivity $= \frac{\text{moles of hydrocinnamaldehyde produced}}{\text{total number of moles of products}} \times 100$. Mole Ratio of cinnamaldehyde: glycerol = 1:10 with glycerol volume kept constant, and in excess, catalyst amount = 0.05, HCAL is hydro-cinnamaldehyde.

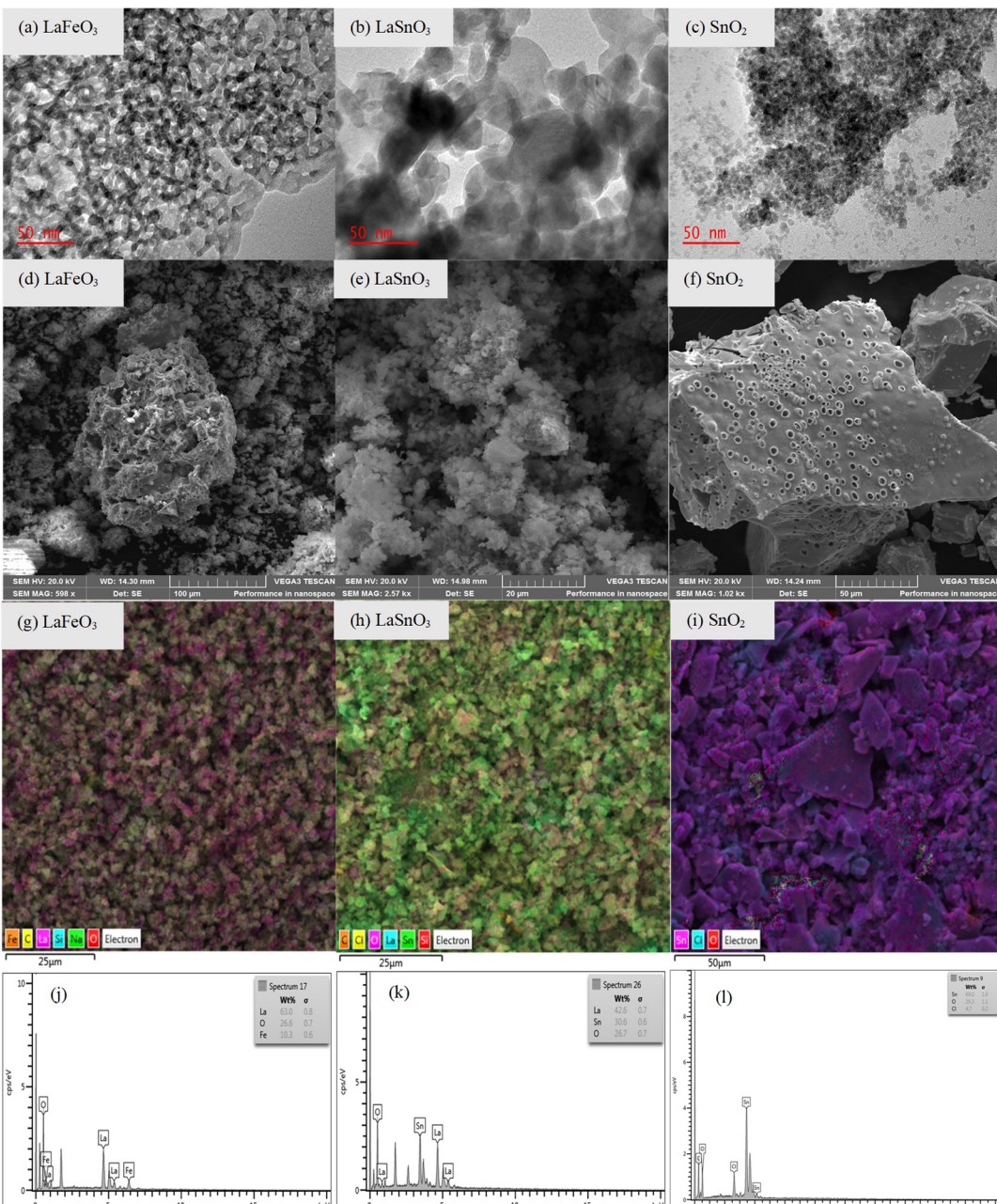

**Figure 3.** TEM images of the synthesized perovskites (**a**–**c**); their corresponding SEM images (**d**–**f**); elemental mapping images (**g**–**i**); and the EDS spectra (**j**–**l**).

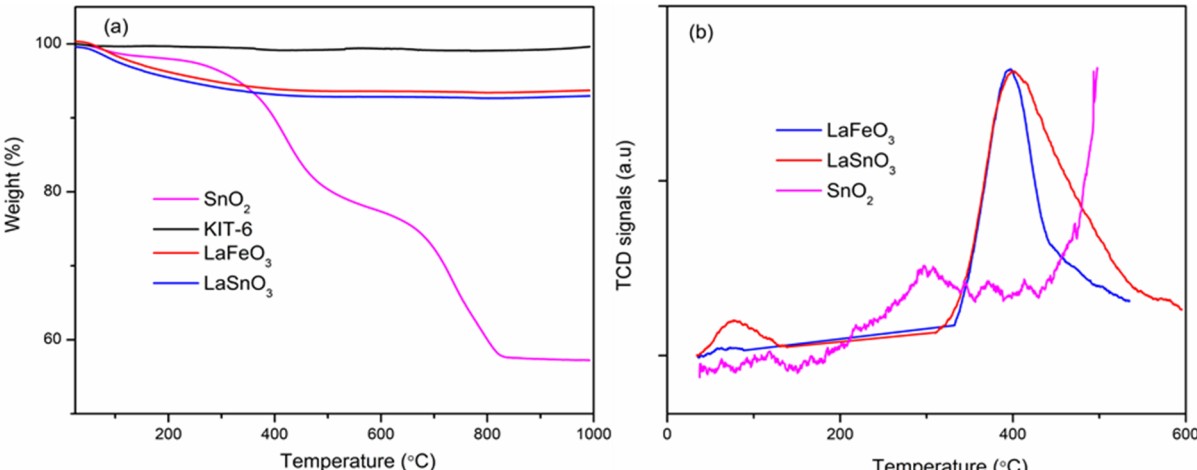

**Figure 4.** (**a**) Thermograms of the synthesized catalyst and the template; (**b**) their corresponding $CO_2$-TPD profiles to determine basic sites. The $LaSnO_3$ thermogram vertically translated by 0.26%.

### 2.2. Catalytic Performance

2.2.1. Reaction Optimization

Increasing the catalyst surface area in the reactor proved that the reaction is a surface reaction. The substrate conversion increased with increasing surface area. The rapid increase in conversion at lower catalyst loading indicates a surface reaction governed by the active catalyst sites. This region of increasing conversions with an increase in surface area is termed the kinetic zone. In contrast, the leveling-off in conversion at high catalyst loading is pronounced and can be attributed to severe mass-transport limitation. Figure 5a shows different reaction zones as a function of catalyst loading.

Once the correct amount of the catalyst needed to avoid the possibility of performing the reaction in mass-transport limited conditions is determined, other reaction parameters can be optimized. In the ratio of substrate to sacrificial alcohol, the glycerol did not impact the percentage conversion of the substrate (see Figure 6a). Following this, in the time and temperature variations, an initial increase in conversions was observed up to 120 °C. Beyond this point, there was no significant change observed in the conversions, as illustrated in Figure 6b. The increase in temperature increased the kinetic energy for the reaction to occur and influenced the substrate adsorption and product desorption, respectively. Similarly, a gradual increase in conversions was observed up to 3 h with leveling-off thereafter (see Figure 5b). A maximum temperature of 120 °C and a reaction time of 3 h were found to be the optimum conditions.

The formation of multiple products during the catalytic transfer hydrogenation of cinnamaldehyde is an issue that has been persistent for some time. With cinnamaldehyde, the formation of cinnamyl alcohol is favored because the $\alpha$ position is more thermodynamically favored [15]. Our perovskite catalytic systems favored hydrogenation at the alpha position and the beta site to form cinnamyl alcohol and hydro-cinnamaldehyde. Although all the catalysts showed good catalytic activity of up to 99% conversions, the selectivity was not maximum. A notable distinction in selectivity trends is the high selectivity towards hydro-cinnamaldehyde by the Fe-containing perovskite.

In comparison, the Sn-containing catalysts favor the formation of cinnamyl alcohol. This trend can be attributed to the basicity and acidity of the catalysts. The more basic catalyst would result in high conversions due to the acceptance of protons from the hydrogen donor to its surface. However, the acidity would increase the selectivity of hydro-cinnamaldehyde because there is higher stabilization of the carbonyl compound. In addition, there seems to be a gradual increase in selectivity towards hydro-cinnamaldehyde with time up to 3 h. Thus, the optimal reaction time of 3 h corresponds to the highest selectivity towards hydro-cinnamaldehyde alcohol. Table 2 shows the catalytic activity of the synthesized materials and selectivity trends.

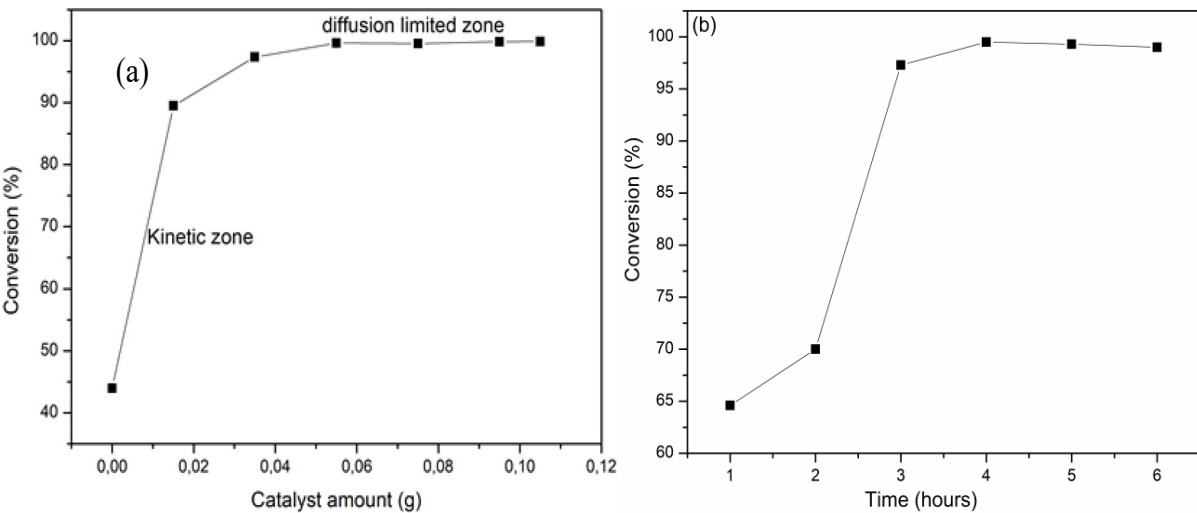

**Figure 5.** Dependence of furfural percentage conversion on (**a**) catalyst amount and (**b**) reaction time.

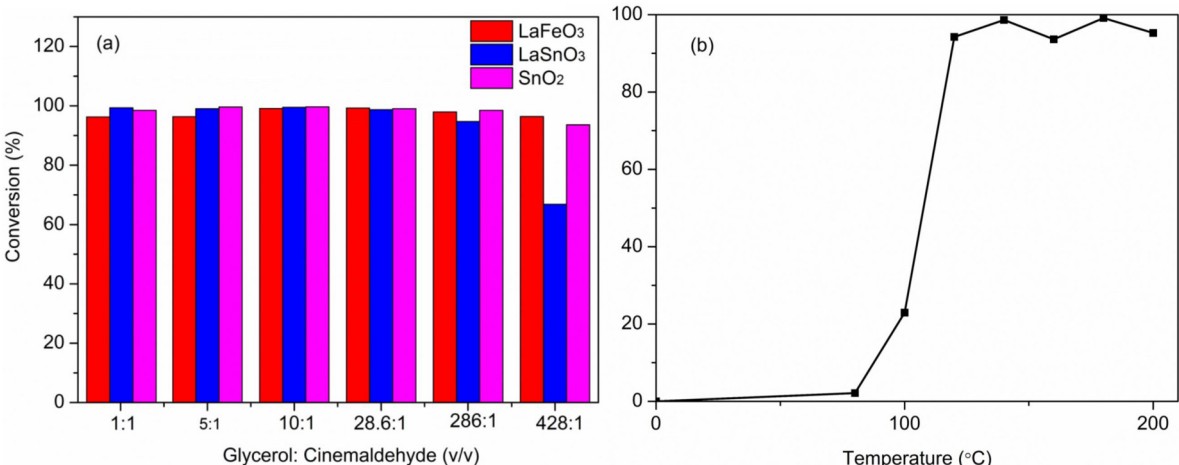

**Figure 6.** Optimization of reaction parameters (**a**) substrate and sacrificial alcohol mole ratio variation and (**b**) temperature variation.

### 2.2.2. Recyclability Tests

To ensure the sustainability, environmental safety, and cost-efficiency of the catalysts, it is vital to recover them after numerous catalytic cycles. The catalysts showed good conversions up to five catalytic cycles, as shown in Figure 7d,e. There were no significant changes in the morphology of the materials, as shown in Figure 7a–c. Furthermore, there was no major loss of catalyst mass, with the maximum loss being 15%, as shown in Table S1. The lowest conversion obtained was 79%. Since there were no major changes in the starting material's percentage conversion, the catalyst integrities are validated as the reaction conditions do not readily deactivate them. This integrity is evidenced by the insignificant change in the normalized mass conversion shown in Figure 7e.

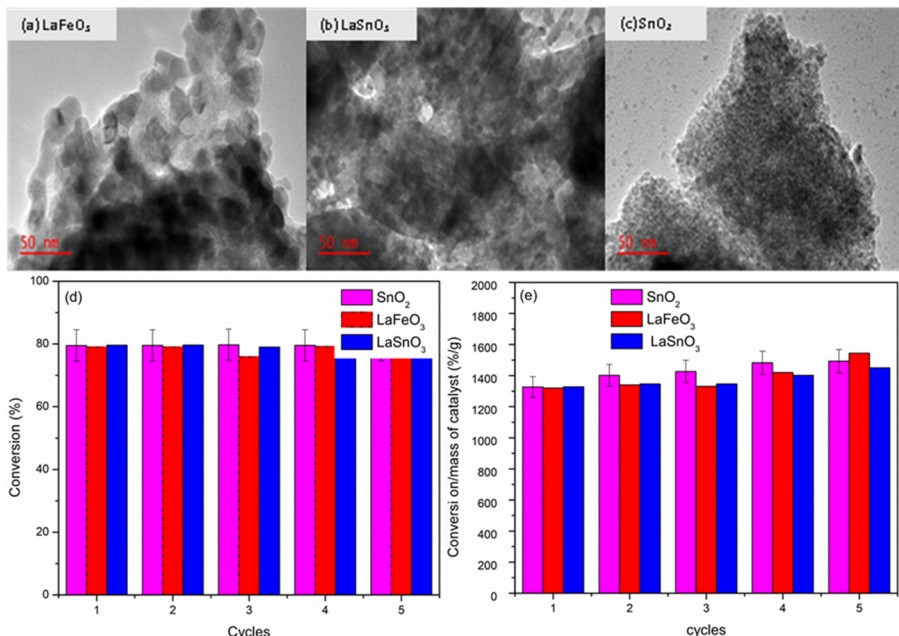

**Figure 7.** TEM image of the spent (**a**) LaFeO$_3$; (**b**) LaSnO$_3$; (**c**) SnO$_2$; (**d**) percentage conversions per cycle; and (**e**) normalised conversions.

### 2.3. Proposed Reaction Mechanism

The efficiency of the catalyst has been reported to be dependent on the basicity of the catalyst. However, both La and Fe oxides were previously active in CTH reactions, with low selectivity due to their alterable oxidation states (Fe$^{3+}$/Fe$^{4+}$) and weak acidity. Better conversions are expected when using perovskites such as LaFeO$_3$ due to the synergy of La and Fe cations. On the other hand, secondary alcohols have been proven to release more hydrogens than primary alcohols [45]. The secondary alcohol moiety of glycerol is believed to be the position from which hydrogen is abstracted first, leading to a ketone moiety on the central carbon (Scheme 2).

**Scheme 2.** The conversion of glycerol to dihydroxyacetone and diglycerol during the CTH reaction.

Furthermore, the availability of enough active sites on the perovskite surface enables the adsorption of glycerol molecules on adjacent sites, triggering an etherification of the two adjacent adsorbed glycerol molecules. It is important to note that the formation of such molecules has been encountered in heterogeneous catalysis. The dimerization reaction usually occurs in the presence of acid/base catalysts in solvent-free conditions [46,47]. The diglycerol molecules can be formed either in cyclic, linear, or branched form depending on the catalyst type, temperature, and reaction time [48]. The demand for the linear diglycerol formed in this work has increased by more than 50% from 2012 to 2022 [47]. The formation of two products in one pot renders the La- and Sn- perovskites versatile catalysts. Scheme 2 shows the conversion of glycerol after the hydrogenation reaction is complete. Schemes 3 and 4 illustrate the products of catalytic transfer hydrogenation and the proposed mechanisms for the formation of the products.

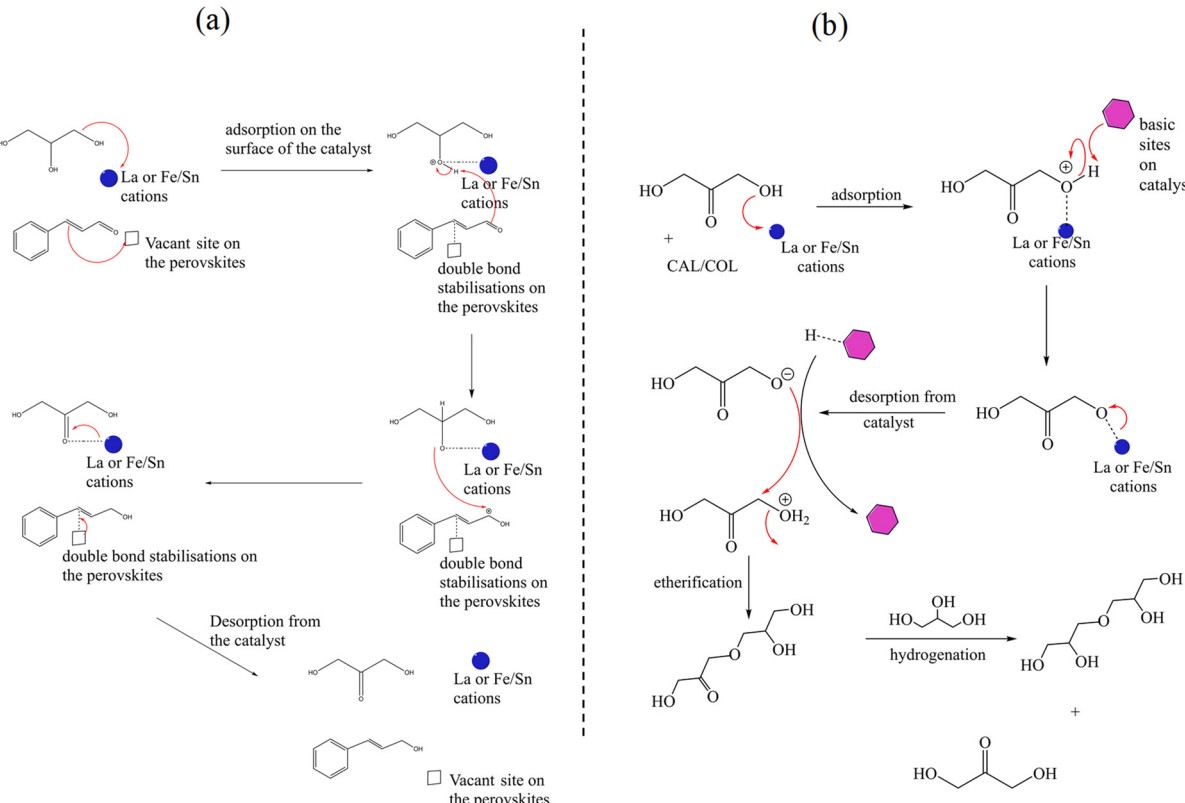

**Scheme 3.** Proposed mechanism showing glycerol transformations during (**a**) the formation of cinnamyl alcohol from cinnamaldehyde and (**b**) the formation of diglycerol.

The mechanism proposes that the hydroxyl group of the glycerol adsorbs to the cationic centers of the perovskite, either the $La^{3+}$ or the $Fe^{3+/4+}$. At that time, the hydrogen atom of the hydroxyl group interacts with the adjacent oxygen anions on the perovskite's lattice structure (Schemes 3 and 4). The mode of adsorption of the cinnamaldehyde molecule on the substrates determines the product distribution. Since both were formed, we propose the adsorption of cinnamaldehyde to be a dual-mode where it adsorbs via both the carbonyl (Scheme 4) and alkene moieties (Scheme 3). The mechanism was adopted from a study conducted by Ping et al. with the variation in the etherification of glycerol and the incorporation of the use of vacant sites from the catalysts [11]. The sacrificial alcohol also adsorbs on the surface, and once it is near another absorbed glycerol molecule, the etherification reaction occurs, Scheme 4b. From Scheme 3a, the formation of cinnamyl alcohol is depicted. It is important to note that the same principle applies for hydrocinnamaldehyde with the double bond attacked instead of the carbonyl compound (see Scheme 4a). A control reaction was run to determine the formation of the diglycerol from two glycerol molecules. It was determined that the dimerization of glycerol to form diglycerol through etherification was significantly low, with 22.8% conversion and a selectivity of 1.2% towards diglycerol and 98.8% towards dihydroxyacetone. Hence, we propose that most of the diglyceride was formed using the mechanism in Scheme 3b instead of Scheme 4b.

**Scheme 4.** Proposed mechanism showing glycerol transformations during (**a**) the formation of hydro-cinnamaldehyde from cinnamaldehyde and (**b**) glycerol etherification to diglycerol.

The GC-MS confirmed the formation of the products and bi-products, as shown in Figure S1 in ESI. As per the GC-MS library, the hydro-cinnamaldehyde, cinnamyl-alcohol, and diglycerol were matched by 77%, 79%, and 78% respectively. The diglycerol mass spectra displayed fragmentation patterns with losses of $CH_3CO$ and $C_4H_9O_3$ equivalent to 43 and 105 amu. Hydro-cinnamaldehyde had peak fragmentations of 91, 105, and 78 amu with losses of 43 amu ($CH_2CHO$), the highest. Cinnamyl alcohol had losses of 42 amu corresponding to $C_2H_2O$ the highest. The products were further confirmed by proton $^1$H-NMR spectra (Figure S3 in ESI) for the reactions represented in Table 2. The characteristic peaks for the substrate and products are cinnamaldehyde (C7; 7.6 ppm), cinnamyl alcohol (C9; 4.4 ppm), and hydro-cinnamaldehyde (C7 and C8; 2.8 and 2.9 ppm).

### 3. Experimental

#### 3.1. Utilized Reagents

Most reagents used for this study were purchased from Sigma Aldrich (Johannesburg, South Africa). These are Pluronic, P-123 (Poly (ethylene glycol)-block-poly (propylene

glycol)-block- poly (ethylene glycol), citric acid ($HOC(COOH)(CH_2COOH)_2$) (99.5%), lanthanum(III) nitrate hexahydrate ($La(NO_3)_3 \cdot 6H_2O$) (99.99%), tetraethyl orthosilicate (TEOS) ($\geq$99.0%), nitric acid ($HNO_3$) (70%), iron(III) nitrate nonahydrate ($Fe(NO_3)_3 \cdot 9H_2O$) (98%), internal standard (n-decane), cyclohexane, and standards such as hydro-cinnamaldehyde, cinnamyl alcohol (COL), cinnamaldehyde (CAL) and hydro-cinnamaldehyde, and cinnamyl alcohol (HCAL). Hydrochloric acid (HCl) (32%) and ethanol (99.8%) were purchased from Associated Chemical Enterprise (ACE), (Johannesburg, South Africa). Butanol and sodium hydroxide pellets (NaOH) were purchased from Rochelle Chemicals, (Johannesburg, South Africa) Milli-Q (18 M$\Omega \cdot$cm) water was utilized in all experiments. All chemicals were of analytical grade and used as received.

### 3.2. Catalyst Preparation

#### 3.2.1. Synthesis of the Hard-Template and Perovskites ($LaSnO_3$ and $LaFeO_3$)

This synthesis was performed in two steps, the synthesis of KIT-6 and then the synthesis of the perovskites. The sol-gel method was used to synthesize the mesoporous silica (KIT-6), which was later used as a template to synthesize the perovskites via nanocasting, known as the hard-template method [15]. For the synthesis of the template, exactly 9.00 g of the surfactant (P-123) was dissolved in 330 mL of water and 17.5 mL of HCl under vigorous stirring. After that, 9 mL of the co-surfactant (butanol) was added to the mixture and stirred for an hour at 35 °C. Thereafter, 19.4 mL of the silicon precursor (tetraethyl ortho-silicate; TEOS) was added into the reaction mixture in a volume ratio of 2.16:1 to the butanol. Subsequently, the mixture was left to stir at 35 °C before being aged at 80 °C for 48 h. The resulting white powder was then filtered and dispensed in a mixture of ethanol and HCl in a 50:30 (v/v) ratio and was calcined at 550 °C to remove the surfactants.

The silicon-based soft-template was then used to synthesize the lanthanum-based perovskites using citric acid as an anchor of the two cations, A and B, of the perovskite $ABO_3$ structure [15]. Initially, the citric acid solution was prepared by dissolving 2.60 g of the citric acid into 40 mL of ethanol. The addition of both the La and metal salts (iron(III) nitrate nonahydrate and stannous chloride) was performed for this solution. In a separate beaker, 4.00 g of the KIT-6 template was dispersed in 40 mL deionized water into which the citric acid–metal salts solution was added. The mixture was stirred overnight before the removal of the solvent and dried at 80 °C for 24 h. Thereafter, the product was subjected to heating cycles, calcined first at 500 °C for 4 h and 700 °C for 6 h to form the template containing the perovskites. The template was removed with 2 M hot NaOH. The product was washed with ethanol and water three times and dried overnight at 80 °C.

#### 3.2.2. Synthesis of $SnO_2$

The synthesis of the $SnO_2$ was ensured by following the procedure stated in previous studies [38]. Similar to the synthesis of KIT-6, the appropriate masses and volumes of the surfactant P-123 (20.00 g), co-surfactant (butanol 120 mL), and 40.00 g of $SnCl_2$ were mixed at room temperature until everything dissolved. Thereafter, 13.56 mL of 70% $HNO_3$ was added dropwise to the mixture with continuous stirring over 3 h. Afterward, the gel-like material was placed in an oven to remove the solvent at 120 °C. The resulting solid was subjected to washing cycles with ethanol before heating cycles at 150 °C for 12 h and 350 °C for 6 h, cooling in-between, and removing all the nitrate ions and the surfactants.

### 3.3. Catalyst Characterization

The surface and porosity analysis of the as-synthesized catalysts was done by the Brunauer–Emmett–Teller (BET) method using a Micromeritics ASAP 2460 setup (Micrometrics, Norcross, GA, USA). The determination of the surface area of the catalysts with the aid of the multiple points measured during the analysis was achieved. The analysis was conducted at −196 °C. Both the $N_2$ adsorption and desorption isotherms were obtained. Before analysis, the samples were degassed under a vacuum for 12 h at 90 °C to remove all adsorbed species.

To study the morphology of the catalysts, the samples were dispersed in methanol and sonicated for 25 min. Thereafter, the samples were dispersed onto carbon-coated copper grids in preparation for analysis with transmission electron microscopy (TEM). The Joel-Jem 2100F microscope (Jeol, Tokyo, Japan) equipped with a field emission gun operating at 200 kV was used.

Prior to p-XRD analysis, the samples were pulverized into fine powder for a more effective analysis. The diffraction patterns were measured on a Philips X' Pert Pro p-XRD instrument from (PANalytical, Almelo, EA, Netherlands) using scan speeds of 10°/min, and the data captured was 0.02° resolved. The samples were analyzed using a Cu K $\alpha_1$ radiation source with a wavelength of 1.54 nm. The operating conditions were a voltage and current of 40 kV using 40 mA, respectively. The low angle measurements were conducted between 0.5 and 10°, while the wide-angle was at 10–90° ($2\theta$). Furthermore, the High Score Performance (HSP) software (Ver 4.9, Malvern PANalalytical, WO, UK, 2020) was used to obtain crystallographic information.

A Tescan Vega 3LMH scanning electron microscope (SEM) (Tescan, Kohoutovice, Brno, Czech Republic) was used for further morphological studies after coating the samples using a carbon-coated sputter. The Essence$^{TM}$ computer software (Tescan, Kohoutovice, Brno, Czech Republic) was used for further analysis using a magnification of 20–200 nm. The EDS was also used for the elemental mapping of the catalysts using an Oxford detector, and the obtained results were then compared with those from p-XRD.

The stability of the catalyst was determined by thermogravimetric analysis (TGA). This test was performed using a TA SDT Q600 thermal analyser (TA Instruments, New Castle, DE, USA). The temperature ranged from 25 °C to 1000 °C with a heating rate of 10 °C/min. The analysis was performed in air.

The reducibility of the catalyst was determined using hydrogen temperature-programmed reduction ($H_2$-TPR) on a Micro metrics AutochemII chemisorption analyser (Micrometrics, Norcross, GA, USA). The temperature was ramped from 25 °C to 900 °C at a heating rate of 10 °C/min to record the reduction profiles. The reduction was ensured using a 10:90 ratio of hydrogen and argon. The same instrument was used for the temperature-programmed desorption studies using a mixture of ammonia and helium with a ratio of 10:90 and a carbon dioxide to helium mixture of the same ratio under the same conditions as analysis gases.

### 3.4. Catalytic Studies

All catalytic evaluations were performed on a parallel reaction carousel station equipped with temperature and stirring speed controllers. Into the carousel tubes equipped with magnetic stirrer bars, the required volumes of glycerol, cinnamaldehyde, internal standard (decane), and 0.075 g of the catalysts were added. Various reaction conditions for catalytic investigations such as temperature, catalyst amount, and the ratio of glycerol to cinnamaldehyde were set accordingly. Temperature conditions were varied from 80 to 200 °C. The catalyst amounts were varied from 45 to 95 mg. In addition, the ratio of glycerol to cinnamaldehyde was varied. After completing the reaction, the contents were analyzed using a Shimadzu GC-FID 2010 (Shimadzu, Johannesburg, South Africa) equipped with Restek Rtx-5 capillary column (30 m length and 0.25 μm diameter) and a flame ionization detector.

### 3.5. Recyclability Tests

To evaluate the reusability and integrity of the catalyst, recyclability tests were run using reaction conditions of approximately eighty percent conversion. This test was performed for all the catalysts to evaluate which was the most stable and recovered catalyst. After each run, the reaction mixture was washed with ethanol and centrifuged at 3000 rpm for 15 min prior to drying under a vacuum at 80 °C.

## 4. Conclusions

Perovskites were successfully employed in the catalytic transfer hydrogenation of cinnamaldehyde to produce hydro-cinnamaldehyde and diglycerol in the same pot. The basicity of the catalysts and the surface area was found to impact the hydrogenation of the substrates. Interestingly, the selectivity for hydro-cinnamaldehyde was more than cinnamyl alcohol over the less basic catalyst because of the preferred binding of the carbonyl group compared to the alkene group onto the vacant site. The hydrogen donor was in excess as compared to cinnamaldehyde to drive the reaction forward. The dimerization of glycerol to form diglycerol was attributed more to the reaction of dihydroxyacetone with another glycerol molecule than from two glycerol molecules. The catalysts were stable under the reducing conditions for up to five cycles without significant changes in their structure under these reducing conditions. Overall, $LaFeO_3$ was the best catalyst for the conversion of cinnamaldehyde to hydro cinnamaldehyde, whereas $LaSnO_3$ was optimum for cinnamaldehyde to cinnamyl alcohol. We recommend optimizing the perovskite catalyst's A and B site cations for better selectivities and higher conversions of cinnamaldehyde.

**Supplementary Materials:** The following supporting information can be downloaded at: https://www.mdpi.com/article/10.3390/catal12020241/s1, Figure S1. Mass spectrum of the major products during the cinnamaldehyde catalytic transfer hydrogenation reaction, that is, cinnamyl alcohol, cinnamaldehyde, and diglycerol, respectively. Figure S2. Carbon dioxide-temperature-programmed reduction profiles for (a) $LaFeO_3$, (b) $LaSnO_3$, and (c) $SnO_2$ at various temperature ramping. Figure S3: A stack of $^1$H-NMR from a product mixture of CAL, COL, and HCAL. Table S1. Mass loss and selectivity variations during different reaction cycles for perovskite and $SnO_2$ (Recyclability results).

**Author Contributions:** Conceptualization, R.M. and N.B.; methodology, T.P.M. and N.B.; formal analysis, T.P.M., R.M. and N.B.; investigation, T.P.M.; data curation, T.P.M.; writing—original draft preparation, T.P.M.; writing—review and editing, N.B.; supervision, R.M. and N.B.; project administration, N.B.; funding acquisition, N.B. All authors have read and agreed to the published version of the manuscript.

**Funding:** This research was funded by the National Research Foundation of South Africa (NRF) (UID: (117997)).

**Data Availability Statement:** Data is contained within the article or Supplementary Material.

**Acknowledgments:** We extend our gratitude to and acknowledge the financial support from South African NRF (Grant specific number 117997 and 111710) and the Analytical division of the University of Johannesburg (spectrum).

**Conflicts of Interest:** The authors declare no conflict of interest.

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
