# Peer review of "The Inorganic Perovskite-Catalyzed Transfer Hydrogenation of Cinnamaldehyde Using Glycerol as a Hydrogen Donor"

_catalysts, doi:10.3390/catal12020241_

Round 1
Reviewer 1 Report
This manuscript by author’s projects an overview of catalytic transfer hydrogenation of cinnamaldehyde using glycerol with the aid of Sn and F-based perovskites such as LaFeO3 and LaSnO3 as a heterogeneous catalysts. I support for publication in catalysts, there are some minor modifications needs to be further clarify prior to acceptance.
Comments
- Previous studies (cite) for these conversions as well as compound numberings should be included in the Scheme 1.
- The authors should include at least proton NMR of the converted products and byproducts in the SI file.
- Authors should highlight mass fragmented peaks of the products in the MS spectra.
- The resolution of the image (chem drawings) of the Scheme 3 is very poor (blurred) and must be include with better resolution.
Author Response
We thank the reviewer for the comments and suggestions. We believe the reviewer's contribution really improved the quality of our work.

Reviewer 2 Report
The manuscript by Mabate et al. reports the catalytic properties of perovskite-type oxides in the transfer hydrogenation reaction. Although perovskite-catalyzed transfer hydrogenation is a known process, the authors present an advantage of their process in using glycerol available from renewable sources. The manuscript fits the scope of Catalysts journal and may be considered for publication after taking into account the following remarks:
- In the introduction part, several references on perovskite-catalyzed transfer hydrogenation are given, however, they are not recent. I suggest including a bit more detailed overview of perovskites in transfer hydrogenation in the introduction part and including recent literature references, e.g. 10.1016/j.apcata.2020.117742, 10.1021/acsami.9b00506. The literature data should be then compared with the presented results in the Discussion part.
- It is not clear from the text whether the mechanism proposed in section 2.3 is based solely on literature data or is it different in any aspects.
- In the conclusions, I suggest highlighting the most active/advantageous catalyst among the three oxides studied.
- Text labels in Scheme 3 are unreadable.
Author Response
The comments from the reviewer are clear and insightful. We thanks the reviewer for improving our work.
